# Research Progress on Propylene Preparation by Propane Dehydrogenation

**DOI:** 10.3390/molecules28083594

**Published:** 2023-04-20

**Authors:** Cheng Zuo, Qian Su

**Affiliations:** College of Chemistry & Chemical and Environmental Engineering, Weifang University, Weifang 261000, China; 17854270427@163.com

**Keywords:** propylene, propane, oxygen carrier, chemical looping, oxidative dehydrogenation

## Abstract

At present, the production of propylene falls short of the demand, and, as the global economy grows, the demand for propylene is anticipated to increase even further. As such, there is an urgent requirement to identify a novel method for producing propylene that is both practical and reliable. The primary approaches for preparing propylene are anaerobic and oxidative dehydrogenation, both of which present issues that are challenging to overcome. In contrast, chemical looping oxidative dehydrogenation circumvents the limitations of the aforementioned methods, and the performance of the oxygen carrier cycle in this method is superior and meets the criteria for industrialization. Consequently, there is considerable potential for the development of propylene production by means of chemical looping oxidative dehydrogenation. This paper provides a review of the catalysts and oxygen carriers employed in anaerobic dehydrogenation, oxidative dehydrogenation, and chemical looping oxidative dehydrogenation. Additionally, it outlines current directions and future opportunities for the advancement of oxygen carriers.

## 1. Introduction

Propylene is primarily utilized at room temperature to produce raw materials, such as acrylic acid, acrolein, acrylonitrile, and polypropylene, which are fundamental to the synthesis of plastics, rubber, and fibers. The global demand for propylene is projected to rise in tandem with the development of the social economy. The primary sources of propylene production are steam cracking of refinery gas and catalytic cracking of heavy oils, such as petroleum [1,2,3,4,5]. Nevertheless, the propylene yield in these processes is significantly restricted. The yield ratio of propylene to ethylene in the steam cracking process is 0.4–0.5, while it is only 4.5% in the catalytic cracking process of heavy oil [6,7,8,9,10].

Over the past few years, several technologies have been investigated worldwide to improve propylene yield, such as propane dehydrogenation [11,12], carbon-tetraolefin disproportionation to propylene [13,14,15], methanol to propylene [16,17,18], and catalytic cracking of olefins to increase propylene production [19,20,21]. Among these technologies, propane dehydrogenation has gained considerable attention, and its development potential is vast. Propane dehydrogenation accounts for 4.5% of the total propylene capacity, and is the third-largest source of propylene production globally [22,23,24,25,26]. The primary techniques for propylene preparation are anaerobic dehydrogenation and oxidative dehydrogenation. The former process is associated with high equipment and catalyst costs. In contrast, oxidative dehydrogenation is exothermic, and has lower equipment and catalyst costs compared to conventional anaerobic dehydrogenation. Furthermore, oxidative dehydrogenation has significant potential to address equilibrium conversion limitations and low selectivity. Extensive research has been conducted on the oxidative dehydrogenation of propane, using various gases and solids with the different oxidizing properties as oxidants [27,28,29,30]. The primary gas oxidants used in oxidative dehydrogenation are O_2_, N_2_O, Cl_2_, and CO_2_. However, CO_2_ has garnered more attention from previous researchers, due to its difficulty in oxidizing propylene and propane. Some studies have explored the role of CO_2_ in the oxidative dehydrogenation of propane, but it is challenging to regulate its impact on the reaction at low temperatures. Moreover, the reaction mechanism is not clear, leading to significant product variability. Anaerobic dehydrogenation typically yields low conversion rates (not exceeding 50%) [31,32], but it has higher dehydrogenation efficiency and propylene selectivity. Therefore, anaerobic dehydrogenation has found practical applications. The aerobic dehydrogenation method entails adding an oxidant to the reaction system. Hydrogen, a byproduct of the decomposition of low-carbon alkanes, reacts with the oxidant to produce H_2_O, which can be separated from the reaction by condensation, maintaining a positive reaction direction. The oxidative dehydrogenation of low-carbon alkanes is exothermic, resulting in lower reaction temperatures. The catalyst is not deactivated by high temperatures, enhancing its application value. However, oxidative dehydrogenation has problems with process control, necessitating high catalyst selectivity. Hence, the development of catalysts with high selectivity for target olefins is the focus of current research on the aerobic dehydrogenation of low-carbon alkanes.

While anaerobic dehydrogenation is effective in dehydrogenation, it has several problems, such as poor catalyst cycling performance, severe carbon accumulation, and low propane conversion, limiting the development of industrial propane dehydrogenation process technology. As of September 2022, China imported 1,686,800 tons of propylene, whereas only 38,400 tons were exported [33]. The production was much lower than the demand, making it necessary to identify a new pathway for practical and reliable target olefin production [34]. Chemical looping oxidative dehydrogenation utilizes the hydrogen produced by the dehydrogenation of low-carbon alkanes to combine with lattice oxygen provided by metal oxide (MeO) oxygen supports to generate water, which is separated from the reaction system by condensation. This drives the reaction equilibrium in the direction of a positive reaction, increasing propane conversion. Furthermore, the slow release of lattice oxygen effectively controls the rate of propylene production and enhances propylene selectivity. Moreover, the oxygen support used in chemical looping oxidative dehydrogenation typically exhibits better cycling performance and meets industrial production requirements [35,36,37]. Thus, there is considerable potential for the development of low-carbon olefin production via chemical looping oxidative dehydrogenation.

Although researchers have made many contributions to the propane dehydrogenation, reviews, such as this one, are still necessary to provide direction for future research. In this paper, we will focus on the mechanism and role of catalysts, or oxygen carriers, in propane dehydrogenation reactions.

## 2. Catalysts for Anaerobic Dehydrogenation Reaction

The catalysts utilized in anaerobic dehydrogenation mainly comprise of Pt-Sn and Cr_2_O_3_ catalysts, with other catalysts being less commonly reported. The Pt-Sn catalysts exhibit high catalytic activity, owing to the presence of the noble metal Pt, which also enhances propylene selectivity. Furthermore, the catalysts demonstrate excellent thermal stability and adaptability to a diverse range of reaction conditions, making them industrially viable for over two decades. In contrast, the Cr_2_O_3_ catalysts are inexpensive and readily available, but their application is limited, due to the presence of the heavy metal element Cr, which is harmful to the environment, making them less desirable.

### 2.1. Platinum-Based Catalyst

Platinum (Pt) is a noble metal, often used in the direction of catalyst dehydrogenation. Yu et al. [38] achieved 34.1% propane conversion and 79.2% propylene selectivity using Pt/Al_2_O_3_ catalyst at 576 °C. However, due to the excessive acidic bits of Al_2_O_3_ support, carbon deposition occurred during the reaction, leading to catalyst deactivation. To address this issue, researchers have improved the activity of Pt/Al_2_O_3_ catalysts by adding catalytic agents, or modifying Al_2_O_3_. Various studies have demonstrated that adding Sn significantly improves catalyst activity. For instance, Hien et al. [39] investigated the role of Sn in reducing catalyst Pt/γ-Al_2_O_3_, and found that Sn addition enabled rapid reductive regeneration of Pt, reducing the occurrence of side reactions. Antolini et al. [40] loaded Pt-Sn onto Al_2_O_3_ for propane dehydrogenation and reported that increasing the amount of the active component Sn improved propane conversion and propylene selectivity. The interaction of Sn with Pt produced different types of alloys that modified the defective sites on the catalyst surface, improving Pt dispersion, propane adsorption, and inhibiting the formation of byproducts. Yang et al. [41] investigated the catalytic performance of PtSn catalysts for propane dehydrogenation using first principles calculations and found that the formation of an alloy facilitated the reaction. Vu et al. [42] suggested that the type and stability of PtSn alloy were positively correlated with the activity and stability of the catalytic propane dehydrogenation reaction of this catalyst. Hauser et al. [43] used density function theory (DFT) to study the reaction path of propane dehydrogenation to propylene and found that replacing a Pt atom in the Pt_4_ cluster with a Sn atom to form a PtSn alloy reduced the activation energy of the rate-controlling step, thereby improving propane conversion and propylene selectivity. Sn transfers electrons to Pt atoms, reducing the desorption energy barrier of propylene and carbon precursors, and hindering the adsorption of propylene on Pt atoms, thus reducing the possibility of side reactions, such as hydrogenolysis and coking.

The size of Pt particles also affects propane dehydrogenation. Kumar et al. [44] prepared Pt/SBA-15 catalysts with varying Pt particle sizes for catalytic propane dehydrogenation and found that Pt particles with a particle size of around 3 nm had higher activity. However, the coking rate and amount of coking were also higher. This is due to smaller Pt particles activating the C-C bond, leading to cleavage reactions. 

The results of Nykanen et al. [45] showed that the adsorption energy (0.52 eV) of propylene on the Pt(111) crystal surface is smaller than its energy barrier (0.81 eV) for deep dehydrogenation. While the adsorption energy (0.81 eV) on the Pt(211) crystal surface is larger than its energy barrier (0.54) for deep dehydrogenation. Propylene is prone to deep cracking and coking on the crystalline surface of Pt. Therefore, monometallic Pt-based catalysts have high activity and low selectivity for propylene in the initial stage of the reaction. When the reaction temperature of Pt-based catalysts is high, it is more likely to bring about the sintering problem of Pt nanoparticles. Currently, the Pt-based catalyst activity could be improved by improving the interaction between Pt and the support, in addition to the introduction of metals such as Sn.

#### 2.1.1. Improvement of Support

Different supports can strongly influence catalytic propane dehydrogenation by Pt-based catalysts. Al_2_O_3_ and molecular sieves are the main supports currently used for propane dehydrogenation. Kikuchi et al. [46] and Kobayashi et al. [47] mixed Al_2_O_3_ with MgO, ZnO, and Fe_2_O_3_ to obtain MgO-Al_2_O_3_, ZnO-Al_2_O_3_, and Fe_2_O_3_-Al_2_O_3_ supports, respectively, followed by loading Pt and Sn. Experimental results showed that the Pt-Sn/ZnO-Al_2_O_3_ catalyst for n-butane dehydrogenation was highly effective. In their catalytic propane dehydrogenation study, Vu et al. [48] employed Pt-Sn catalysts that were loaded with La, Ce, and Y-doped Al_2_O_3_ supports. The authors observed that La and Y could form a dispersed phase, whereas Ce formed CeO_2_, due to an agglomeration phenomenon on the catalyst surface. Notably, PtSn/La-Al_2_O_3_ and PtSn/Y-Al_2_O_3_ surfaces formed Pt and Sn alloys, respectively. The catalytic activity of these two catalysts was high, due to the low coking amount and the excellent stability of the alloys.

In contrast to metal oxides, molecular sieves used as supports can reduce the adsorption capacity of propylene and minimize side reactions such as product cracking. In a study conducted by Chen et al. [49], a PtSnNaLa/ZSM-5 catalyst was prepared, and was observed to have a lower amount of carbon deposition compared to the PtSnNaLa/γ-Al_2_O_3_ catalyst during propane dehydrogenation catalysis. After 880 h of continuous reaction, the propane conversion of the PtSnNaLa/ZSM-5 catalyst remained above 30%, while, after 480 h of continuous reaction with the PtSnNaLa/γ-Al_2_O_3_ catalyst, the propane conversion had dropped below 30%. Li et al. [50] prepared Co-doped HZSM-5 catalysts, and it was found that the dehydrogenation reaction rate of propane catalyzed by this catalyst was 12 times higher than that of the HZSM-5 catalyst, and the selectivity of propylene was also high.

#### 2.1.2. Effect of Additives

The addition of metal Sn to the catalyst is also an important factor. Although the Pt-Sn/γ-Al_2_O_3_ catalyst showed a significant improvement in catalytic activity, it still suffered from catalyst coking, which shortened its lifetime [51]. To address this issue, Xia et al. [52] used Mg(Al)O-x supports loaded with active components Pt and In to produce Pt-In/Mg(Al)O-x catalysts. The addition of In regulated the acidity of the catalyst surface, improved the dispersion of Pt, and increased the anti-coking ability of the catalyst. Consequently, the Pt-In/Mg(Al)O-4 catalyst reduced carbon accumulation and prolonged the catalyst’s lifespan. The initial conversion of propane was 66.4%, and the propane conversion after eight reaction cycles still reached 43.5%. Similarly, Zhang et al. [53] added different levels of La to Al_2_O_3_ supports, using the sol-gel method. The best conversion and selectivity of propane were achieved when the mass fraction of La was 1.0%, resulting in 41% propane conversion and 97–98% propylene selectivity. In addition to its effectiveness in the dehydrogenation of propane, the addition of In was also found to be effective in the dehydrogenation of butane. Bocanegra et al. [54] added In to the Pt-Sn system, using MgAl_2_O_4_ as a support for the dehydrogenation of butane, which resulted in high selectivity (95–96%) of butene. During anaerobic dehydrogenation, researchers observed that the competitive adsorption of additives decreased the adsorption of low-carbon olefins, but improved the selectivity of target products produced from the dehydrogenation of low-carbon alkanes.

### 2.2. Cr-Based Catalyst

Cr-based catalysts have gained attention, due to their high catalytic activity and propylene selectivity, as well as their cost-effectiveness compared to noble metal Pt. Also, the better cycling performance of chromium-based catalysts is an important reason for their industrialization. The Cr_2_O_3_/Al_2_O_3_ catalyst, developed by Cabrera et al. [55], demonstrated propane conversion of up to 47% and propylene selectivity above 90% at a reaction temperature of 600 °C and atmospheric pressure. However, carbon deposition and deactivation of the catalyst remain issues that need to be addressed to improve the conversion of propane and selectivity of propylene. Therefore, modifications to Cr-based catalysts are necessary. 

#### Modification of Supports

Kim et al. [56] examined the impact of varying the ratio of Al_2_O_3_ and ZrO_2_ in Cr_2_O_3_ catalyst supports on propylene yield. Their findings showed that the lowest oxygen content of the catalyst was achieved at an Al/Zr ratio of 0.1, resulting in propylene selectivity and yield of 85% and 30%, respectively. The authors speculated that this might be due to the interaction of the active component with the support, leading to the conversion of lattice oxygen to electrophilic oxygen in the catalyst. However, an increase in carbon oxide content (CO_2_ and CO) was observed, leading to a decrease in propylene selectivity. 

It is important to note that Cr is a heavy metal element that poses environmental pollution risks, which greatly limits the widespread industrial use of Cr_2_O_3_.

### 2.3. Introduction of Several Propane Anaerobic Dehydrogenation Industrialization Technologies

Currently, industrial technology for anaerobic propane dehydrogenation mainly consists of the Oleflex process, developed by UP, and the Catofin process, developed by ABB Lummus.

#### 2.3.1. Catofin Process

The Catofin process comprises four stages: propane dehydrogenation to propylene (reaction stage), compression of the reactor discharge (compression stage), and recovery and refining of the product (recovery and refinement stages). The Catofin process employs a CrO_x_/Al_2_O_3_ catalyst, which is cost-effective, and has high cycle times and excellent mechanical properties. The catalyst has a long service life of up to 600 days [21]. 

The main characteristics of the Catofin process [20] are (1) the use of a low-cost non-precious metal catalyst with excellent mechanical properties and high cycle times, (2) high-pressure reaction, requiring the importation of specialized equipment, and (3) easy separation of products.

Figure 1 shows that process diagram of Catofin dehydrogenation unit.

#### 2.3.2. Oleflex Process

The Oleflex process is divided into three parts: the reaction part, the product separation part, and the catalyst regeneration part. The reaction section uses moving bed reactor. Compared with the Catofin process, the catalyst in the reactor is recycled and has a service life of 2 to 7 days.

The Oleflex process employs the Pt catalyst to carry out the dehydrogenation of propane, and the resulting polymeric grade propylene is obtained by separation and distillation in the presence of the catalyst. This reaction does not require the use of hydrogen or water vapor as diluents, resulting in lower energy consumption and operational costs. The Oleflex process is characterized by (1) high operational safety, a small reaction volume, and easy operation, and (2) a lower one-way conversion and a higher sulfur content limitation (not exceeding 100 ppm) compared to the Catofin process. Table 1 shows the comparison of these two process technologies.

Figure 2 shows that Process diagram of Oleflex dehydrogenation unit. Table 1 shows the comparison of the Catofin and Oleflex propane dehydrogenation processes.

The reaction is a strong heat absorption reaction, which requires a large amount of external reaction heat supply. Since the dehydrogenation reaction is an equilibrium reaction, increasing the temperature and decreasing the pressure are beneficial for the dehydrogenation reaction to proceed and obtain a high propane conversion. The temperature of industrial propane dehydrogenation reaction is 500–700 °C. However, the high temperature will promote the occurrence of thermal cracking side reactions, which will also produce some heavy hydrocarbons and form a small amount of coking on the catalyst, thus reducing the reactivity. Therefore, the Oleflex process is more selective for propylene than the Catofin process, due to the cyclic regeneration of the catalyst.

## 3. Catalysts for Oxidative Dehydrogenation Reaction

In contrast to anaerobic dehydrogenation, oxidative dehydrogenation is a highly endothermic reaction that is not limited by thermodynamic equilibrium, thereby increasing propane conversion. However, it is prone to catalyst deactivation, due to carbon deposition. To extend the catalyst lifetime, oxidative dehydrogenation reactions typically require the introduction of a gaseous oxidant. Common oxidants include O_2_, N_2_O, and CO_2_. CO_2_, in particular, has been widely studied in the literature, as it does not deeply oxidize propane or propylene. The function of CO_2_ as an oxidant can be attributed to two factors [57,58]: (1) the reaction CO_2_ + C → 2CO can reduce carbon deposition on the catalyst and improve its stability, and (2) it inhibits the adsorption of olefin products on the catalyst surface, thereby improving propylene selectivity.

### 3.1. Chromium-Based Catalysts

The use of Cr_2_O_3_/Al_2_O_3_ catalysts in oxidative dehydrogenation is also common. However, unlike in anaerobic dehydrogenation, the addition of CO_2_ as an oxidant does not improve propane conversion and propylene selectivity of the Cr_2_O_3_/Al_2_O_3_ catalyst. Its only function is to extend the catalyst’s lifetime. Therefore, researchers have explored the use of molecular sieves to modulate the physicochemical properties of Cr_2_O_3_. Michorczyk et al. [59] loaded Cr_2_O_3_ onto a MCM-41 molecular sieve and obtained 34.9% propane conversion and 88.5% propylene selectivity at 550 °C. Zhang et al. [60] loaded Cr_2_O_3_ onto SBA-15, ZrO_2_, and ZrO_2_/SBA-15 supports and found that the Cr_2_O_3_/SBA-15 catalyst displayed excellent catalytic activity, with 24.2% propane conversion and 83.9% propylene selectivity at 600 °C.

### 3.2. Vanadium-Based Catalysts

V_2_O_5_ is an acidic oxide that exhibits high catalytic activity, but low propylene selectivity. Therefore, V_2_O_5_ is often loaded onto suitable supports to improve propylene selectivity for propane dehydrogenation. The appropriate support can decrease the deep dehydrogenation capability of V_2_O_5_ and enhance the selectivity of propylene. The catalytic activity center in vanadium-based catalysts is VO_x_ [61]. Vanadium oxide with high coordination numbers can deeply oxidize propane, whereas highly dispersed tetrahedral VO_4_ provides limited lattice oxygen for propane dehydrogenation. By controlling the release rate of lattice oxygen, selectivity of propylene can be improved. A balanced ratio of acidic and basic sites on the catalyst surface is the key to improving the conversion of propane and propylene selectivity [62]. A more acidic surface activates propane more strongly, improving the propane conversion. On the other hand, the product propylene has a greater electron cloud density compared to propane, and it is more basic. Therefore, a more alkaline surface facilitates propylene desorption and Improves selectivity [63,64,65,66]. During the preparation of vanadium-based catalysts, it is crucial to control the vanadium content. Exceeding the theoretical monolayer of vanadium content results in the appearance of octahedral V_2_O_5_ crystalline phases, with different polymerization deformations on the catalyst surface [67,68,69,70]. Therefore, it is essential to disperse tetrahedral VO_4_ as much as possible on the catalyst surface to reduce the occurrence of crystalline phase V_2_O_5_. Hossain et al. prepared vanadium-based CaO-γ-Al_2_O_3_ supports for the oxidative dehydrogenation of propane [71]. They achieved 25.5% conversion of propane and 94.2% selectivity of propylene at 640 °C, and the most active catalysts were obtained at a mass ratio of CaO to γ-Al_2_O_3_ of 1:1.

### 3.3. Gallium-Based Catalysts

Gallium-based catalysts have also been utilized for propane dehydrogenation, along with chromium-based and vanadium-based catalysts. Ga_2_O_3_ catalysts operate via a heterolysis process, which is distinct from the mechanism of the Cr system. The reaction mechanism is illustrated in Figure 3 [72,73,74,75,76].

Xu et al. [77] and Ren et al. [78] discovered that the impact of CO_2_ oxidation on propane dehydrogenation was evident when the reaction rate of (3c) was slow and the reaction rate of (3d) was fast. On the other hand, the presence of CO_2_ had little effect on the reaction when the reaction rate of (3c) was fast and the reaction rate of (3d) was slow. However, CO_2_ had an inhibitory effect on propane dehydrogenation, as it had to compete with propane for the basic sites on the catalyst surface, which hindered the adsorption of propane on the catalyst surface. When the rate of reaction (3d) was very slow, the conversion of propane and propylene yield decreased with the increase of CO_2_ concentration.

## 4. The Process of Chemical Looping Oxidative Dehydrogenation

The anaerobic dehydrogenation method has drawbacks, such as the non-recyclability of the catalyst, and being constrained by thermodynamic equilibrium, resulting in low conversion rates. On the other hand, the aerobic dehydrogenation method has issues, such as difficulty controlling the degree of reaction, especially when using CO_2_ as the oxidant, leading to varying reaction products. Chemical looping technology uses an oxygen carrier that can be regenerated and slowly releases lattice oxygen to control the degree of reaction, thereby improving the thermodynamic irreversibility of traditional dehydrogenation reactions. Chemical looping oxidative dehydrogenation overcomes the limitations of both anaerobic and oxidative dehydrogenation methods, and has the potential to significantly improve the conversion of low-carbon alkanes and selectivity of low-carbon olefins [79,80,81].

Chemical looping oxidative dehydrogenation involves oxidation and reduction reactions based on the oxygen carrier’s reaction type in two reactors. In the dehydrogenation reactor, the oxygen carrier is used for the dehydrogenation reaction with propane, and is then regenerated with air, releasing heat in the oxidation reactor. During the reaction, the products of low carbon alkanes after dehydrogenation (H_2_) combine with the metal oxide oxygen carriers’ lattice oxygen to form water, which is removed from the reaction system by condensation, promoting the reaction equilibrium to proceed in the positive reaction direction, thus increasing the conversion rate of low carbon alkanes. The lattice oxygen in the oxygen carrier can be gradually released under specific conditions, controlling the reaction’s course, which contributes to enhancing the selectivity of propylene. After the reduction of the oxygen carrier product in the dehydrogenation reactor, it enters the air reactor for oxidation with oxygen to complete the regeneration process. The process flowchart is presented in Figure 4.

### 4.1. Monometallic Active Oxygen Carriers

Ghamdi et al. [82] investigated catalysts with varying vanadium content (5%, 7%, 10% wt.%), loaded onto γ-Al_2_O_3_ for chemical looping oxidative dehydrogenation reactions. They achieved a maximum propylene selectivity of 85.94% at a propylene conversion of 11.73%. However, the VO_x_/γ-Al_2_O_3_ catalysts had a limited number of cycles, with a maximum of 10 cycles throughout the reaction. This was likely due to the accumulation of V_2_O_5_ crystal structures on the catalyst surface as the number of cycles increased, which decreased the propylene yield. In monometallic oxygen carriers, lattice oxygen is released rapidly, resulting in deep propane oxidation during the chemical looping oxidative dehydrogenation of propane. Additional CO_2_ is often required to provide an oxygen source for the reaction system. Loading appropriate metal oxides onto catalysts to control the rate of lattice oxygen release and regulate the migration or evolution of the released lattice oxygen from the catalyst surface could reduce the selectivity of CO_x_ and extend the catalyst’s lifespan. Wu et al. [83] compared the activity of Ga-based, Mo-based, and V-based oxygen carriers for propane dehydrogenation at 540 °C, 615 °C, and 650 °C, respectively. The results showed that the V-based oxygen carrier had the best catalytic activity. Meanwhile, the optimum reaction temperature for propane dehydrogenation was also investigated to be 615 °C.

### 4.2. Bimetallic or Polymetallic Composite Oxygen Carriers

Fukudome et al. [84,85] achieved higher concentrations of isolated VO_x_ species by incorporating them into the SiO_2_ framework using alkoxy exchange between metal alcohol oxygen compounds and polyethylene glycols. It was observed that SiO_2_-doped VO_x_ exhibited higher selectivity for propylene than VO_x_ loaded onto SiO_2_. Gao et al. [86] used polymetallic composite oxygen carriers (La*_x_*Sr_2−*x*_FeO_4−*δ*_) to dehydrogenate ethane into ethylene, with a high yield of 51.6%, and the number of cycles of the oxygen carriers reached up to 30. Thus, there is a need to develop oxygen carriers that can last for more cycles in the future.

Appropriate bimetallic or even polymetallic oxides can release lattice oxygen slowly, which is more effective in controlling the rate of propane dehydrogenation for propylene production than anaerobic dehydrogenation and gaseous oxidants. Moreover, metal oxides can inhibit the conversion of lattice oxygen (O^2–^) to electrophilic oxygen (O_2_^–^) and reduce the formation of oxides (e.g., CO*_x_*), thereby improving the conversion of propane and the selectivity of propylene [87,88,89]. Additionally, the short reaction time of chemical looping oxidative dehydrogenation, ranging from 20 s to 8 min, makes it difficult to study the reaction mechanism of propane dehydrogenation. For industrial promotion, further research is needed to develop oxygen carriers with high oxygen loading capacity and high propane conversion with propylene selectivity.

## 5. Conclusions and Prospects

(1)The current methods for propylene production are anaerobic and oxidative dehydrogenation. The anaerobic method has been used for many years, but is expensive, due to high equipment and catalyst costs. The oxidative dehydrogenation method is cheaper, but the extent of CO_2_ influence on the reaction is difficult to control at certain temperatures, and the reaction mechanism is still unclear, resulting in variable product yields.(2)In contrast, chemical looping oxidative dehydrogenation resolves the drawbacks of the previous methods. Lattice oxygen release can be controlled by appropriate bimetallic or polymetallic oxides, replacing molecular oxygen. This effectively controls the reaction rate of propane dehydrogenation to produce propylene, and improves the conversion of propane with high selectivity for propylene, compared to oxygen-free dehydrogenation and gas oxidant methods.(3)The future of chemical looping oxidative dehydrogenation for industrial applications requires the development of multi-component coupled composite oxygen carriers with high oxygen loading, extended cycle life, and high propylene yield.

## Figures and Tables

**Figure 1 molecules-28-03594-f001:**
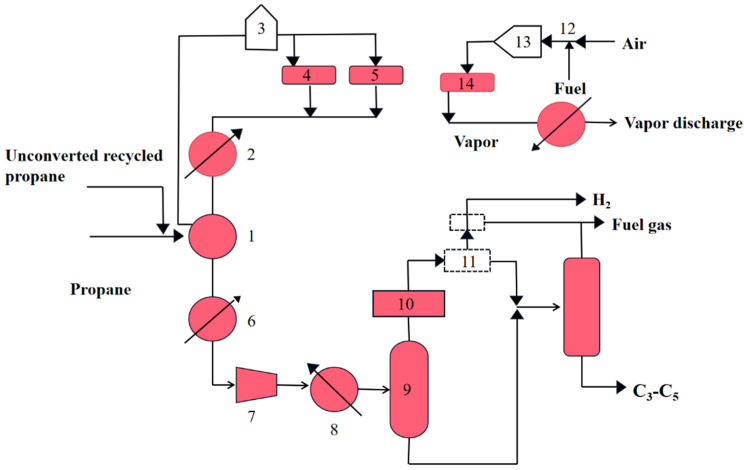
Process diagram of Catofin dehydrogenation unit. (1) Heat exchanger; (2) Steam generator; (3) Heating furnace; (4) Purge section reactor; (5) Process section reactor; (6) Cooler; (7) Compressor; (8) Air cooling; (9) Flash tank; (10) Dryer; (11) Cold box; (12) Gasifier; (13) Heater; (14) Regeneration section reactor.

**Figure 2 molecules-28-03594-f002:**
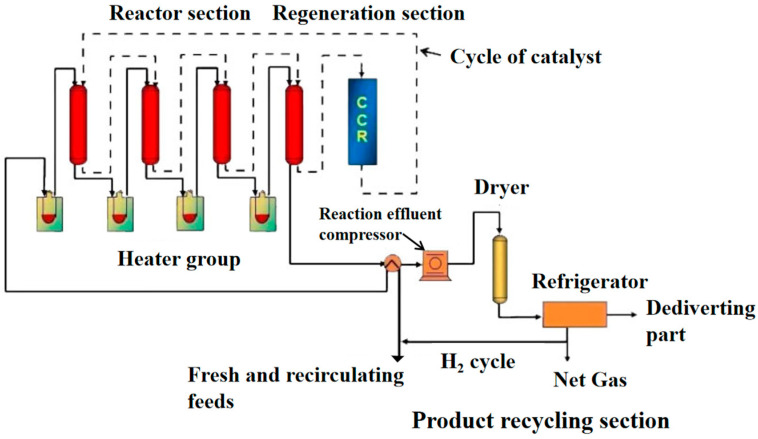
Process diagram of Oleflex dehydrogenation unit.

**Figure 3 molecules-28-03594-f003:**
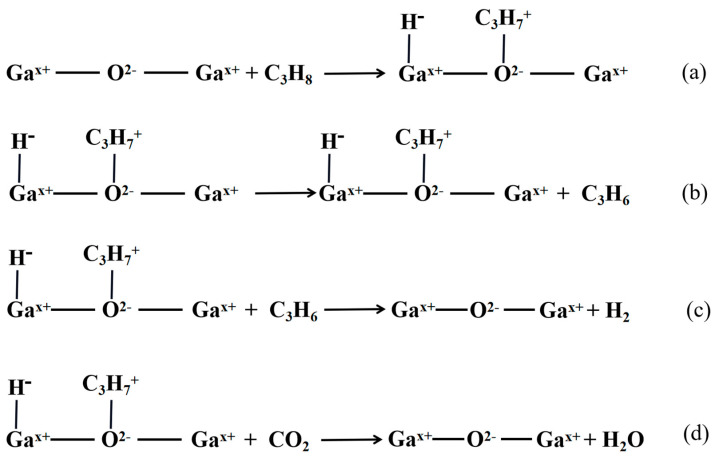
Reaction mechanism diagram of Ga_2_O_3_ catalyst with propane (**a**–**d**).

**Figure 4 molecules-28-03594-f004:**
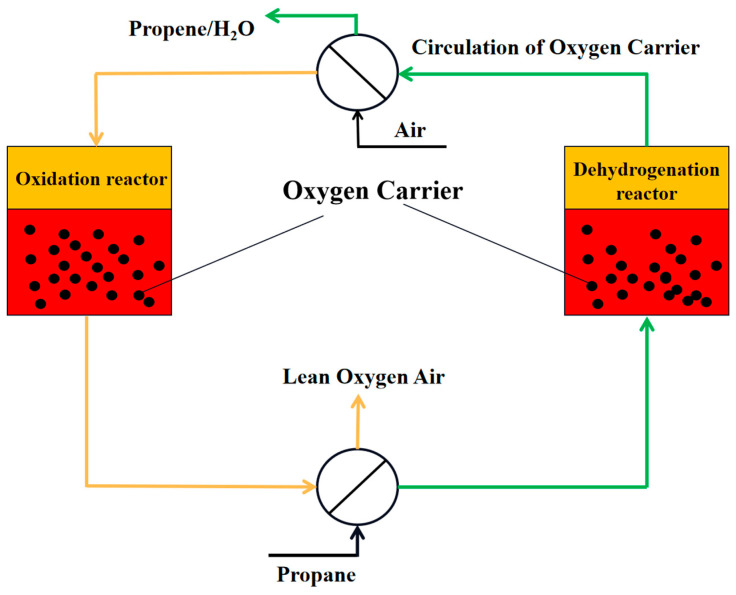
The process flow diagram of chemical looping oxidative dehydrogenation.

**Table 1 molecules-28-03594-t001:** Comparison of Catofin and Oleflex propane dehydrogenation process technologies.

Projects	Process Technology	
	Catofin Process	Oleflex Process
Technology exporter	ABB Lummus	UOP
Reactor type	Fixed Bed	Moving Bed
Total number of reactors	5	3~4
Catalyst	CrO_x_/Al_2_O_3_	Pt-Sn/Al_2_O_3_
Cycle regeneration time	15~30 min	2~7 d
Temperature/°C	600–700	550~620
Pressure/Mpa	0.3~0.5	2~3
Diluent	-	H_2_
Propane conversion	48~65	80~88
Propylene selectivity	25	89~91

## Data Availability

Not applicable.

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
