# Peer review of "Research Progress on Propylene Preparation by Propane Dehydrogenation"

_molecules, 2023, doi:10.3390/molecules28083594_

Round 1

Reviewer 1 Report

1. 3.1 Chromium-based catalysts. Why are there full name of element for Chromium and Vanadium but not for Pt?

2. An introduction for Catofin Process is needed in the beginning of 2.3.1. Also for the Oleflex process.

3. A sentence should be added in the last of introduction section to address the main focus of your review. Please refer to other review articles.

4. Please summarize two current industrial processes for propane and summarize them in a table.

5. Compared to the noble metal Pt, Cr was cheap and easy to obtain as a noble metal-free catalyst. Compared with other noble metal-free catalyst, it has more high catalytic activity with the selectivity of propylene. The reason should not originate from the price, but from the scientific viewpoint.

English language and style are minor spell check required. English should be improved. 

Author Response

Dear Editors and Reviewers,

Thanks for your comments concerning our manuscript entitled “Research progress on propylene preparation by propane dehydrogenation” (manuscript ID 2361628). Those comments are all valuable and helpful for revising and improving our paper. We have carefully studied the comments and made corrections based on your comments. Modified portions are highlighted in red in the revised version. As suggested, we improved the manuscripts by improving the writing and adding more data and analyses. The revised manuscript is responded to, point by point.

The primary corrections in the paper and the responses to the reviewer’s comments are as follows.

Thank you and best regards.

Yours sincerely,

Corresponding author:      [email protected].

  1. 1 Chromium-based catalysts.Why are there full name of element for Chromium and Vanadium but not for Pt?

Response: Thanks for the comment.

2.1 Platinum-based catalyst. Platinum(Pt) is a noble metal, often used in the direction of catalyst dehydrogenation.

  1. An introduction for Catofin Process is needed in the beginning of 2.3.1. Also for the Oleflex process.

Response: Thanks for the comment. The Oleflex process is divided into three parts: the reaction part, the product separation part and the catalyst regeneration part. The reaction section uses moving bed reactor. Compared with the Catofin process, the catalyst in the reactor is recycled and has a service life of 2 to 7 days.

  1. A sentence should be added in the last of introduction section to address the main focus of your review. Please refer to other review articles.

Response: Thanks for the comment. Although researchers have made many contributions to the propane dehydrogenation. This review such as this one are still necessary to provide direction for future research. In this paper, we will focus on the mechanism and role of catalysts or oxygen carriers in the propane dehydrogenation reactions.

  1. Please summarize twocurrent industrial processes for propane and summarize them in a table.

Response: Thanks for the comment. Table 1 shows the comparison of these two process technologies.

Table 1. Comparison of Catofin and Oleflex propane dehydrogenation process technologies.

Projects

Process Technology

Catofin process

Oleflex process

Technology Exporter

ABB Lummus

UOP

Reactor Type

Fixed Bed

Moving Bed

Total number of reactors

5

3~4

Catalyst

CrOx/Al2O3

Pt-Sn/Al2O3

Cycle regeneration time

15~30 min

2~7 d

Temperature/oC

600-700

550~620

Pressure/Mpa

0.3~0.5

2~3

Diluent

-

H2

Propane conversion

48~65

80~88

Propylene selectivity

25

89~91

  1. Compared to the noble metal Pt, Cr was cheap and easy to obtain as a noble metal-free catalyst. Compared with other noble metal-free catalyst, it has more high catalytic activity with the selectivity of propylene. The reason should not originate from the price, but from the scientific viewpoint.

Response: Thanks for the comment. Also, the better cycling performance of chromium-based catalysts is an important reason for their industrialization. 

Reviewer 2 Report

Propylene is an important building block for enormous petrochemicals. Propane dehydrogenation (PDH) is an industrial technology for direct propylene production which has received extensive attention in recent years. In this work, the authors provide a review of the catalysts and oxygen carriers employed in anaerobic dehydrogenation, oxidative dehydrogenation, and chemical looping oxidative dehydrogenation of propan.

         This work is acceptable for publication in Molecules after major revisions.

There still some questions need the authors to answer and further improve this review.

         My suggestions and questions are listed below.

-              A more detailed analysis of the advantages and disadvantages of the considered methods for producing propylene, their comparison and thermodynamics needs to be done.

-              The active sites, reaction pathways and deactivation mechanisms of PDH over metals and metal oxides as well as their dependent factors also needs to be analysed and discussed for all propylene preparation techniques.

-              The authors also need to present more examples of monometallic reactive oxygen carriers (section 4.1).

-              Papers only up to 2019 are present in the References. Much has been published in the last three years. I would recommend adding new materials, for example:

Zhou J, Zhang Y, Liu H, et al. Enhanced performance for propane dehydrogenation through Pt clusters alloying with copper in zeolite. Nano Research, 2023, https://doi.org/10.1007/s12274-022-5317-z

Dan Zhao, Xinxin Tian, Dmitry E. Doronkin, et al. In situ formation of ZnOx species for efficientpropane dehydrogenation. Nature, Vol 599, 11 November 2021 https://doi.org/10.1038/s41586-021-03923-3

Sai Chen, Xin Chang, et al. Propane dehydrogenation: catalyst development, new chemistry, and emerging technologies. Chem. Soc. Rev., 2021, 50, 3315 DOI: 10.1039/d0cs00814a

-              Some references are not formatted correctly (see the template), the years of several references are not in bold.

Author Response

Dear Editors and Reviewers,

Thanks for your comments concerning our manuscript entitled “Research progress on propylene preparation by propane dehydrogenation” (manuscript ID 2361628). Those comments are all valuable and helpful for revising and improving our paper. We have carefully studied the comments and made corrections based on your comments. Modified portions are highlighted in red in the revised version. As suggested, we improved the manuscripts by improving the writing and adding more data and analyses. The revised manuscript is responded to, point by point.

The primary corrections in the paper and the responses to the reviewer’s comments are as follows.

Thank you and best regards.

Yours sincerely,

Corresponding author:      [email protected].

  1. A more detailed analysis of the advantages and disadvantages of the considered methods for producing propylene, their comparison and thermodynamics needs to be done.

Response: Thanks for the comment. The reaction is a strong heat absorption reaction, which requires a large amount of external reaction heat supply. Since the dehydrogenation reaction is an equilibrium reaction, increasing the temperature and decreasing the pressure are beneficial for the dehydrogenation reaction to proceed and obtain a high propane conversion rate. The temperature of industrial propane dehydrogenation reaction is 500-700°C. However, the high temperature will promote the occurrence of thermal cracking side reactions, which will also produce some heavy hydrocarbons and form a small amount of coking on the catalyst, thus reducing the reactivity. Therefore, the Oleflex process is more selective for propylene than the Catofin process due to the cyclic regeneration of the catalyst. 

Table 1. Comparison of Catofin and Oleflex propane dehydrogenation process technologies.

Projects

Process Technology

Catofin process

Oleflex process

Technology Exporter

ABB Lummus

UOP

Reactor Type

Fixed Bed

Moving Bed

Total number of reactors

5

3~4

Catalyst

CrOx/Al2O3

Pt-Sn/Al2O3

Cycle regeneration time

15~30 min

2~7 d

Temperature/oC

600-700

550~620

Pressure/Mpa

0.3~0.5

2~3

Diluent

-

H2

Propane conversion

48~65

80~88

Propylene selectivity

25

89~91

  1. The active sites, reaction pathways and deactivation mechanisms of PDH over metals and metal oxides as well as their dependent factors also needs to be analysed and discussed for all propylene preparation techniques.

Response: Thanks for the comment. The results of Nykanen et al. [89] showed that the adsorption energy (0.52 eV) of propylene on the Pt(111) crystal surface is smaller than its energy barrier (0.81 eV) for deep dehydrogenation. While, the adsorption energy (0.81 eV) on the Pt(211) crystal surface is larger than its energy barrier (0.54) for deep dehydrogenation. Propylene is prone to deep cracking and coking on the crystalline surface of Pt. Therefore, monometallic Pt-based catalysts have high activity and low selectivity for propylene in the initial stage of the reaction. When the reaction temperature of Pt-based catalysts is high, it is more likely to bring about the sintering problem of Pt nanoparticles. Currently, the Pt-based catalyst activity could be improved by improving the interaction between Pt and the support, in addition to the introduction of metals such as Sn.

The use of metal oxides for propane dehydrogenation is still in the laboratory research stage and many reaction mechanisms are still unclear. There is no uniform conclusion yet.

[89] Nykanen, L.; Honkala K. Selectivity in Propene Dehydrogenation on Pt and Pt3Sn Surfaces

from First Principles. ACS Catal. 2013, 3, 3026-3030.

  1. The authors also need to present more examples of monometallic reactive oxygen carriers (section 4.1).

Response: Thanks for the comment. Wu et al. [82] compared the activity of Ga-based, Mo-based and V-based oxygen carriers for propane dehydrogenation at 540 oC, 615 oC and 650 oC, respectively. The results showed that the V-based oxygen carrier had the best catalytic activity. Meanwhile, the optimum reaction temperature for propane dehydrogenation was also investigated to be 615 oC.

[82] Wu,T.W.;  Yu, Q.B.;  Hou, L.M.;   Duan, W.J.; Wang, K.;  Qin, Q. Selecting suitable oxygen carriers for chemical looping oxidative dehydrogenation of propane by thermodynamic method. J. Therm. Anal. Calorim. 2020, 140,1837–1843.

  1. Papers only up to 2019 are present in the References. Much has been published in the last three years. I would recommend adding new materials, for example:

Zhou J, Zhang Y, Liu H, et al. Enhanced performance for propane dehydrogenation through Pt clusters alloying with copper in zeolite. Nano Research, 2023, https://doi.org/10.1007/s12274-022-5317-z

Dan Zhao, Xinxin Tian, Dmitry E. Doronkin, et al. In situ formation of ZnOx species for efficientpropane dehydrogenation. Nature, Vol 599, 11 November 2021 https://doi.org/10.1038/s41586-021-03923-3

Sai Chen, Xin Chang, et al. Propane dehydrogenation: catalyst development, new chemistry, and emerging technologies. Chem. Soc. Rev., 2021, 50, 3315 DOI: 10.1039/d0cs00814a

Response: Thanks for the comment.

[3] Wang, H.M.; Chen, Y.; Yan, X.; Lang, W.Z.; Guo, Y.J. Cr doped mesoporous silica spheres for propane dehydrogenation in the presence of CO2: Effect of Cr adding time in sol-gel process. Micropor. Mesopor. Mat. 2019, 284, 69-77.

[5] Chen, S.; Chang, X.; Sun, GD.; Zhang, TT.; Xu, YY.; Wang, Y.; Pei, C.L.; Gong, J.L. Propane dehydrogenation: catalyst development, new chemistry, and emerging technologies. Chem. Soc. Rev. 2021, 50, 3315.

[10] Zhao, D.; Tian, X.X , Dmitry E. In situ formation of ZnOx species for efficientpropane dehydrogenation. Nature2021, DOI:10.1038/s41586-021-03923-3.

[11] Zhang, H.J.; Wan, H.; Zhao, Y.; Wang, W.Q. Effect of chlorine elimination from Pt-Sn catalyst on the behavior of hydrocarbon reconstruction in propane dehydrogenation. Catal. Today. 2023, DOI: 10.1007/s12274-022-5317-z.

[50] Zhou, J.; Zhang, Y.; Liu, H.; Xiong, C.; Hu, P.; Wang, H.; Chen, S.W.; Ji, H.B. Enhanced performance for propane dehydrogenation through Pt clusters alloying with copper in zeolite. Nano Res. 2023, DOI: 10.1007/s12274-022-5317-z.

  1. Some references are not formatted correctly (see the template), the years of several references are not in bold.

Response: Thanks for the comment. The formatting of the references has been corrected and the year of the references has been bolded.

Round 2

Reviewer 2 Report

The manuscript has been sufficiently improved to warrant publication in Molecules